# RPM: GENERALIZABLE MULTI-AGENT POLICIES FOR MULTI-AGENT REINFORCEMENT LEARNING

**Wei Qiu**[‡][*] **Xiao Ma**[†] **Bo An**[‡] **Svetlana Obraztsova**[‡] **Shuicheng Yan**[†] **Zhongwen Xu**[†][✉]
[‡]Nanyang Technological University  [†]Sea AI Lab
qiuw0008@e.ntu.edu.sg    zhongwen.s.xu@gmail.com

## ABSTRACT

Despite the recent advancement in multi-agent reinforcement learning (MARL), the MARL agents easily overfit the training environment and perform poorly in evaluation scenarios where other agents behave differently. Obtaining generalizable policies for MARL agents is thus necessary but challenging mainly due to complex multi-agent interactions. In this work, we model the MARL problem with Markov Games and propose a simple yet effective method, called ranked policy memory (RPM), *i.e.*, to maintain a look-up memory of policies to achieve good generalizability. The main idea of RPM is to train MARL policies via gathering massive multi-agent interaction data. In particular, we first rank each agent's policies by its training episode return, *i.e.*, the episode return of each agent in the training environment; we then save the ranked policies in the memory; when an episode starts, each agent can randomly select a policy from the RPM as the behavior policy. Each agent uses the behavior policy to gather multi-agent interaction data for MARL training. This innovative self-play framework guarantees the diversity of multi-agent interaction in the training data. Experimental results on Melting Pot demonstrate that RPM enables MARL agents to interact with unseen agents in multi-agent generalization evaluation scenarios and complete given tasks. It significantly boosts the performance up to 818% on average.

## 1 INTRODUCTION

In Multi-Agent Reinforcement Learning (MARL) (Yang & Wang, 2020), each agent acts decentrally and interacts with other agents to complete given tasks or achieve specified goals via reinforcement learning (RL) (Sutton & Barto, 2018). In recent years, much progress has been achieved in MARL research (Vinyals et al., 2019; Jaderberg et al., 2019; Perolat et al., 2022). However, the MARL agents trained with current methods tend to suffer poor generalizability (Hupkes et al., 2020) in the new environments. The generalizability issue is critical to real-world MARL applications (Leibo et al., 2021), but is mostly neglected in current research.

In this work, we aim to train MARL agents that can adapt to new scenarios where other agents' policies are unseen during training. We illustrate a two-agent hunting game as an example in Fig. 1. The game's objective for two agents is to catch the stag together, as one agent acting alone cannot catch the stag and risks being killed. They may perform well in evaluation scenarios similar to the training environment, as shown in Fig. 1 (a) and (b), respectively, but when evaluated in scenarios different from the training ones, these agents often fail. As shown in Fig. 1 (c), the learning agent (called the focal agent following (Leibo et al., 2021)) is supposed to work together with the other agent (called the background agent also following (Leibo et al., 2021)) that is pre-trained and can capture the hare and the stag. In this case, the focal agent would fail to capture the stag without help from its teammate. The teammate of the focal agent may be tempted to catch the hare alone and not cooperate, or may only choose to cooperate with the focal agent after capturing the hare. Thus, the focal agent should adapt to their teammate's behavior to catch the stag. However, the policy of the background agent is unseen to the focal agent during training. Therefore, without generalization, the agents trained as Fig. 1 (left) cannot achieve an optimal policy in the new evaluation scenario.

---

[*]Wei Qiu did the work while interning at Sea AI Lab. [✉] Corresponding author.

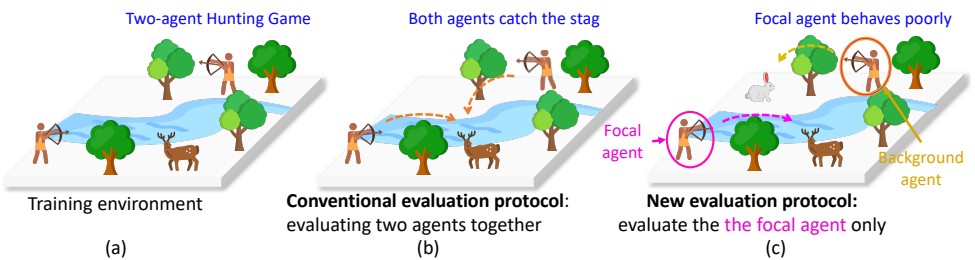

Figure 1: Two-Agent Hunting Game. (a) Training environment. Two agents (hunters) hunt in the environment. (b) After training in the training environment, all agents behave cooperatively to capture the stag. (c) In the new evaluation scenario, one agent is picked as the focal agent (in the magenta circle) and paired with a pre-trained agent (in the brown circle) that behaves in different ways to evaluate the performance of the selected agent. In conclusion, the conventional evaluation protocol fails to evaluate such behavior and current MARL methods easily fail to learn the optimal policy due to the lack of diversified multi-agent interaction data during training.

Inspired by the fact that human learning is often accelerated by interacting with individuals of diverse skills and experiences (Meltzoff et al., 2009; Tomasello, 2010), we propose a novel method aimed at improving the generalization of MARL through the collection of diverse multi-agent interactions. Concretely, we first model the MARL problem with Markov Games (Littman, 1994) and then propose a simple yet effective method called ranked policy memory (RPM) to attain generalizable policies. The core idea of RPM is to maintain a look-up memory of policies during training for the agents. In particular, we first evaluate the trained agents' policies after each training update. We then rank the trained agents' policies by the training episode returns and save them in the memory. In this way, we obtain various levels, *i.e.*, the performance of the policies. When starting an episode, the agent can access the memory and load a randomly sampled policy to replace the current behavior policy. The new ensemble of policies enables the agents in self-play to collect diversified experiences in the training environment. These diversified experiences contain many novel multi-agent interactions that can enhance the extrapolation capacity of MARL, thus boosting the generalization performance. We note that an easy extension by incorporating different behavior properties as the keys in RPM could potentially further enrich the generalization but it is left for future work.

We implement RPM on top of the state-of-the-art MARL algorithm MAPPO (Yu et al., 2021). To verify its effectiveness, we conduct large-scale experiments with the Melting Pot (Leibo et al., 2021), which is a well-recognized benchmark for MARL generalization evaluation. The experiment results demonstrate that RPM significantly boosts the performance of generalized social behaviors up to 818% on average and outperforms many baselines in a variety of multi-agent generalization evaluation scenarios. Our code, pictorial examples, videos and experimental results are available at this link: https://sites.google.com/view/rpm-iclr2023/.

## 2 PRELIMINARIES

**Markov Games.** We consider the Markov Games (Littman, 1994) represented by a tuple $\mathcal{G} = \langle \mathcal{N}, \mathcal{S}, \mathcal{A}, \mathcal{O}, P, R, \gamma, \rho \rangle$. $\mathcal{N}$ is a set of agents with the size $|\mathcal{N}| = N$; $\mathcal{S}$ is a set of states; $\mathcal{A} = \times_{i=1}^{N} \mathcal{A}_i$ is a set of joint actions with $\mathcal{A}_i$ denoting the set of actions for an agent $i$; $\mathcal{O} = \times_{i=1}^{N} \mathcal{O}_i$ is the observation set, with $\mathcal{O}_i$ denoting the observation set of the agent $i$; $P : \mathcal{S} \times \mathcal{A} \to \mathcal{S}$ is the transition function and $R = \times_{i=1}^{N} r_i$ is the reward function where $r_i : \mathcal{S} \times \mathcal{A} \to \mathbb{R}$ specifies the reward for the agent $i$ given the state and the joint action; $\gamma$ is the discount factor; the initial states are determined by a distribution $\rho : \mathcal{S} \to [0, 1]$. Given a state $s \in \mathcal{S}$, each agent $i \in \mathcal{N}$ chooses its action $u_i$ and obtains the reward $r(s, \boldsymbol{u})$ with the private observation $o_i \in \mathcal{O}_i$, where $\boldsymbol{u} = \{u_i\}_{i=1}^{N}$ is the joint action. The joint policy of agents is denoted as $\boldsymbol{\pi_\theta} = \{\pi_{\theta_i}\}_{i=1}^{N}$ where $\pi_{\theta_i} : \mathcal{S} \times \mathcal{A}_i \to [0, 1]$ is the policy for the agent $i$. The objective of each agent is to maximize its total expected return $R_i = \sum_{t=0}^{\infty} \gamma^t r_i^t$.

**Multi-Agent RL.** In MARL, multiple agents act in the multi-agent systems to maximize their respective returns with RL. Each agent's policy $\pi_i$ is optimized by maximizing the following objective:

$$\mathcal{J}(\pi_i) \triangleq \mathbb{E}_{s_{0:\infty} \sim \rho_{\mathcal{G}}^{0:\infty}, a_{0:\infty}^i \sim \pi_i} \left[ \sum_{t=0}^{\infty} \gamma^t r_t^i \right],$$

where $\mathcal{J}(\pi_i)$ is a performance measure for policy gradient RL methods (Williams, 1992; Lillicrap et al., 2016; Fujimoto et al., 2018). Each policy's Q value $Q_i$ is optimized by minimizing the following regression loss (Mnih et al., 2015) with TD-learning (Sutton, 1984):

$$\mathcal{L}(\theta_i) \triangleq \mathbb{E}_{\mathcal{D}' \sim \mathcal{D}} \left[ \left( y_t^i - Q_{\theta_i}^i \left( \boldsymbol{s}_t, \boldsymbol{u}_t, s_t^i, u_t^i \right) \right)^2 \right],$$

Figure 2: An example of our formulation. **Left:** All six agents' policies are trained with the MARL. **Right:** Two agents with policies $\pi_{\phi_1}$ and $\pi_{\phi_2}$ are picked as background agents, and the rest of the four agents (with new indices) are focal agents to be evaluated. The focal and the background agents constitute the evaluation scenario.

where $y_t^i = r_t^i + \gamma \max_{\boldsymbol{u}'} Q_{\bar{\theta}_i}^i \left( \boldsymbol{s}_{t+1}, \boldsymbol{u}', s_t^i, u^{i,\prime} \right)$. $\theta_i$ are the parameters of the agents. $\bar{\theta}_i$ is the parameter of the target $Q^i$ and periodically copied from $\theta$. $\mathcal{D}'$ is a sample from the replay buffer $\mathcal{D}$.

## 3 PROBLEM FORMULATION

We introduce the formulation of MARL for training and evaluation in our problem. Our goal is to improve generalizabiliby of MARL policies in scenarios where policies of agents or opponents are unseen during training while the physical environment is unchanged. Following Leibo et al. (2021), the training environment is defined as *substrate*. Each substrate is an $N$-agent partially observable Markov game $\mathcal{G}$. Each agent optimizes its policy $\pi_{\theta_i}$ via the following protocol.

**Definition 1** (Multi-Agent Training). There are $N$ agents act in the substrate, which is denoted as $\mathcal{G}$. Each agent receives partial environmental observation not known to other agents and aims to optimizes its policy $\pi_{\theta_i}$ by optimizing its accumulated rewards: $\sum_{t=0}^{\infty} \gamma^t r_t^i$. The performance of the joint policy $\boldsymbol{\pi_\theta} = \{\pi_{\theta_i}\}_{i=1}^N$ is measured by the mean individual return: $\bar{R}(\boldsymbol{\pi_\theta}) = \frac{1}{N} \sum_{i=1}^N R(\pi_{\theta_i}; \mathcal{G})$. $R(\pi_{\theta_i}; \mathcal{G})$ measures the episode return of policy $\pi_{\theta_i}$ in game $\mathcal{G}$ for agent $i$.

In order to evaluate the trained MARL policies in evaluation scenario $\mathcal{G}'$, we follow the evaluation protocol defined by Leibo et al. (2021):

**Definition 2** (Multi-Agent Evaluation). There are $M$ $(1 \leq M \leq N-1)$ focal agents that are selected from $N$ agents. The focal agents are agents to be evaluated in evaluation scenarios. They are paired with $N - M$ background agents whose policies $\boldsymbol{\pi_\phi} = \{\pi_{\phi_j}\}_{j=1}^{N-M}$ were pre-trained with pseudo rewards in the same physical environment where the policies $\boldsymbol{\pi_\theta}$ are trained. To measure the generalized performance in evaluation scenarios, we use the mean individual return of focal agents as the performance measure: $\bar{R}(\{\pi_\theta\}_{i=1}^M) = \frac{1}{M} \sum_{i=1}^M R(\pi_{\theta_i}; \mathcal{G}')$.

We show an example of our formulation in Fig. 2. Note that the focal agents cannot utilise the interaction data collected during evaluation to train or finetune their policies. Without training the policies of focal agents with the collected trajectories during evaluation, the focal agents should behave adaptively to interact with the background agents to complete challenging multi-agent tasks. It is also worth noting that the ad-hoc team building (Stone & Kraus, 2010; Gu et al., 2021) is different from our formulation both in the training and evaluation. We discuss the differences in the related works section (Paragraph 3, Sec. 7).

In MARL, the focal agents need adaptively interact with background agents to complete given tasks. Formally, we define the objective for optimizing performance of the focal agents without exploiting their trajectories in the evaluation scenario for training the policies $\{\pi_{\theta_j}\}_{j=1}^M$:

$$\max \mathcal{J}(\{\pi_{\theta_j}\}_{j=1}^M) \triangleq \max \mathbb{E}_{s_{0:\infty} \sim \rho_{\mathcal{G}'}^{0:\infty}, a_{0:\infty}^j \sim \{\pi_{\theta_j}\}_{j=1}^M} \left[ \sum_{t=0}^{\infty} \gamma^t \frac{1}{M} \sum_{j=1}^M r_t^j \,\middle|\, \mathcal{G}' \right]. \tag{1}$$

## 4 RANKED POLICY MEMORY

To improve the generalization of MARL, agents in the substrate must cover as much as multi-agent interactions, *i.e.*, data, that resemble the unseen multi-agent interactions in the evaluation scenario. However, current training paradigms, like independent learning (Tampuu et al., 2017) and centralized

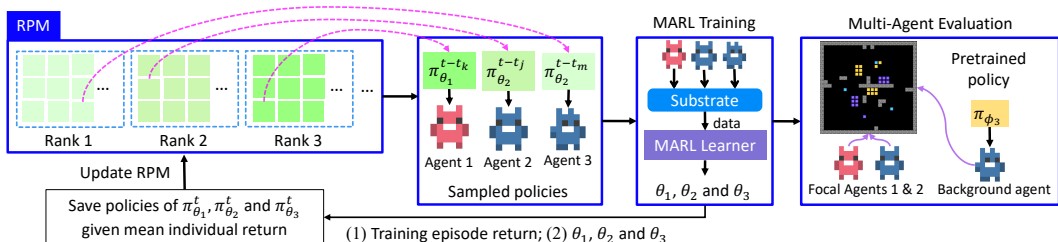

Figure 3: The workflow of RPM for a three-agent substrate. In the workflow, there are three agents in the substrate. Agent 3 is the background agent. Agents 1 and 2 are focal agents.

training and decentralized execution (CTDE) (Oliehoek et al., 2008), cannot give diversified multi-agent interactions, as the agents' policies are trained at the same pace. To this end, we propose a Ranked Policy Memory (RPM) method to provide diversified multi-agent behaviors.

**RPM Building & Updating.** We denote an RPM with $\Psi$, which consists of $|R_{\max}|$ entries, *i.e.*, ranks, where $|R_{\max}|$ is the maximum training episode return (the episode return in the substrate). When an agent is acting in the substrate, it will receive the training episode return $R$ of all agents with policies $\{\pi_\theta^i\}_{i=1}^N$. Then $\{\pi_\theta^i\}_{i=1}^N$ are saved into $\Psi$ by appending agents' policies into the corresponding memory slot, $\Psi[r_e].\text{add}(\{\pi_e^i\}_{i=1}^N)$. To avoid there being too many entries in the policy memory caused by continuous episode return values, we discretize the training episode return. Each discretized entry $\kappa$ covers a range of $[\kappa, \kappa + \psi)$, where $\psi > 0$ and it can be either an integer or a float number. For the training episode return $R$, the corresponding entry $\kappa$ can be calculated by:

$$\kappa = \begin{cases} \lfloor R/\psi \rfloor \times \mathbf{1}\{(R \bmod \psi) \neq 0\} \times \psi, & \text{if } R \geq 0, \\ \lfloor R/\psi \rfloor \times \psi, & \text{otherwise.} \end{cases} \quad (2)$$

where $\mathbf{1}\{\cdot\}$ is the indicator function, and $\lfloor \cdot \rfloor$ is the floor function. Intuitively, discretizing $R$ saves memory and memorize policies of similar performance in to the same rank. Therefore, diversified policies can be saved to be sampled for agents.

**RPM Sampling.** The memory $\Psi$ stores diversified policies with different levels of performance. We can sample various policies of different ranks and assign each policy to each agent in the substrate to collect multi-agent trajectories for training. These diversified multi-agent trajectories can resemble trajectories generated by the interaction with agents possessing unknown policies in the evaluation scenario. At the beginning of an episode, we first randomly sample $N$ keys with replacement and then randomly sample one policy for each key from the corresponding list. All agents' policies will be replaced with the newly sampled policies for multi-agent interactions in the substrate, thus generating diversified multi-agent trajectories.

**The Workflow of RPM.** We showcase an example of the workflow of RPM in Fig. 3. There are three agents in training. Agents sample policies from RPM. Then all agents collect data in the substrate for training. The training episode return is then used to update RPM. During evaluation, agents 1 and 2 are selected as focal agents and agent 3 is selected as the background agent. We present the pseudo-code of MARL training with RPM in Algorithm 1. In Lines 4-5, the $\pi_{\theta_b}$ is updated by sampling policies from RPM. Then, new trajectories of $\mathcal{D}$ are collected in Line 6. $\pi_\theta$ is trained in

---

**Algorithm 1:** MARL with RPM

1  **Input**: Initialize $\boldsymbol{\pi}_\theta$, $\Psi$, $\mathcal{D}$, $\mathcal{G}$ and $\mathcal{G}'$;
2  **Input**: Initialize behavior policy $\boldsymbol{\pi}_{\theta_b} \leftarrow \boldsymbol{\pi}_\theta$;
3  **for** each update **do**
4     **if** `RPM sampling` **then**
5        $\boldsymbol{\pi}_{\theta_b} \leftarrow \text{SamplingRPM}(\Psi)$;
6     $\mathcal{D} \leftarrow \text{GatherTrajectories}(\boldsymbol{\pi}_{\theta_b}, \mathcal{G})$;
7     $\boldsymbol{\pi}_\theta \leftarrow \text{MARLTrainig}(\boldsymbol{\pi}_\theta, \mathcal{D})$;
8     $\Psi \leftarrow \text{UpdateRPM}(\boldsymbol{\pi}_\theta, \Psi, \mathcal{G})$;
9     $\bar{R} \leftarrow \text{Evaluate}(\boldsymbol{\pi}_\theta, \mathcal{G}')$;
10    $\boldsymbol{\pi}_{\theta_b} \leftarrow \boldsymbol{\pi}_\theta$;
11 **Output**: $\boldsymbol{\pi}_\theta$.

---

Line 7 with MARL method by using the newly collected trajecotries and $\pi_{\theta_b}$ is updated with the newly updated $\pi_\theta$. RPM is updated in Line 8. After that, the performance of $\pi_\theta$ is evaluated in the evaluation scenario $\mathcal{G}'$ and the evaluation score $\bar{R}$ is returned in Line 9.

**Discussion.** RPM leverages agents' previously trained models in substrates to cover as many patterns of multi-agent interactions as possible to achieve generalization of MARL agents when paired with agents with unseen policies in evaluation scenarios. It uses the self-play framework for data collection. Self-play (Brown, 1951; Heinrich et al., 2015; Silver et al., 2018; Baker et al., 2019) maintains a memory of the opponent's previous policies for acquiring equilibria. RPM differs from other self-play methods in four aspects: (i) self-play utilizes agent's previous policies to create fictitious opponents

when the real opponents are not available. By playing with the fictitious opponents, many fictitious data are generated for training the agents. In RPM, agents load their previous policies to diversify the multi-agent interactions, such as multi-agent coordination and social dilemmas, and all agents' policies are trained by utilizing the diversified multi-agent data. (ii) Self-play does not maintain explicit ranks for policies while RPM maintains ranks of policies. (iii) Self-play was not introduced for generalization of MARL while RPM aims to improve the generalization of MARL. In Sec. 6, we also present the evaluation results of a self-play method.

## 5 MARL TRAINING

We incorporate RPM into the MARL training pipeline. We take MAPPO (Yu et al., 2021) for instantiating our method, which is a multi-agent variant of PPO (Schulman et al., 2017) and outperforms many MARL methods (Rashid et al., 2018; 2020; Wang et al., 2021a) in various complex multi-agent domains. In MAPPO, a central critic is maintained for utilizing the concealed information of agents to boost multi-agent learning due to non-stationarity. RPM introduces a novel method for agents to collect experiences/trajectories $\boldsymbol{\tau} = \{\tau_i\}_{i=1}^N$. Each agent optimizes the following objective:

$$\mathcal{J}(\theta_i) = \mathbb{E}\left[\min\left(\eta_i^t\left(\theta_i^t\right) \cdot A_i^t, \texttt{clip}\left(\eta_i^t\left(\theta_i^t\right), 1-\epsilon, 1+\epsilon\right) \cdot A_i^t\right)\right], \tag{3}$$

where $\eta_i^t(\theta_i^t) = \frac{\pi_{\theta_i^t}(u_i^t|\tau_i^t)}{\pi_{\theta_i^{\text{old}}}(u_i^t|\tau_i^t)}$ denotes the important sampling weight. The $\texttt{clip}(\cdot)$ clips the values of $\theta^i$ that are outside the range $[1-\epsilon, 1+\epsilon]$ and $\epsilon$ is a hyperparameter. $A_i^t$ is a generalized advantage estimator (GAE) (Schulman et al., 2015). To optimize the central critic $V_\psi(\{o_i^t, u_i^t\}_{i=1}^N)$, we mix agents' observation-action pairs and output an $N$-head vector where each value corresponds to the agent's value:

$$\mathcal{L}(\psi) := \mathbb{E}_{\mathcal{D}' \sim \mathcal{D}}\left[\left(y_t - V_{\bar{\psi}}(\{o_i^t, u_i^t\}_{i=1}^N)\right)^2\right], \tag{4}$$

where $y_t = \left[\sum_{l=0}^{k-1} \gamma^l r_i^{t+l} + \gamma^k V_{\bar{\psi}}(\{o_i^{t+k}, u_i^{t+k}\}_{i=1}^N)[i]\right]_{i=1}^N$ is a vector of $k$-step returns, and $\mathcal{D}'$ is a sample from the replay buffer $\mathcal{D}$. In complex scenarios, *e.g.*, Melting Pot, with an agent's observation as input, its action would not impact other agents' return, since the global states contain redundant information that deteriorates multi-agent learning. We present the whole training process, the network architectures of the agent and the central critic in Appx. D.

## 6 EXPERIMENTS

In this section, to verify the effectiveness of RPM in improving the generalization of MARL, we conduct extensive experiments on Melting Pot and present the empirical results. We first introduce Melting Pot, baselines and experiment setups. Then we present the main results of RPM. To demonstrate that $\psi$ is important for RPM, we conducted ablation studies. We finally showcase a case study to visualize RPM. To sum up, we answer the following questions: **Q1**: Is RPM effective in boosting the generalization performance of MARL agents? **Q2**: How does the value of $\psi$ impact RPM training? **Q3**: Does RPM gather diversified policies and trajectories?

### 6.1 EXPERIMENTAL SETUP

**Melting Pot.** To demonstrate that RPM enables MARL agents to learn generalizable behaviors, we carry out extensive experiments on DeepMind's Melting Pot (Leibo et al., 2021). Melting Pot is a suite of testbeds for the generalization of MARL methods. It proposes a novel evaluation pipeline for the evaluation of the MARL method in various domains. That is, all MARL agents are trained in the substrate; during evaluation, some agents are selected as the focal agents and the rest agents become the background agents (pre-trained policies of MARL models will be loaded); the evaluation scenarios share the same physical properties as the substrates. Melting Pot environments possess many properties, such as temporal coordination and

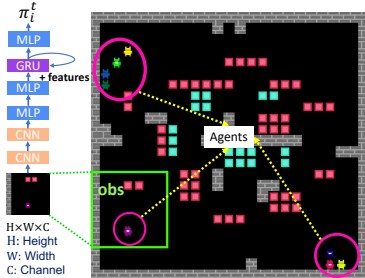

Figure 4: The green box to the lower left shows the agent's observation.

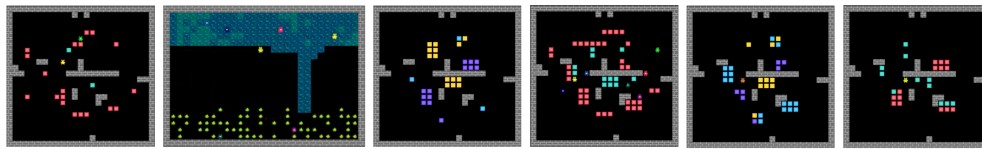

Figure 5: Melting Pot environments. More information can be found in Appx. A.

Table 1: Properties of Melting Pot environments. The first column shows the properties and the first row lists environments. ✓ mark indicates the environment possessing the corresponding property while ✗ mark stands for the environment that does not own the corresponding property. Refer to Appx. A for more information.

| | Stag Hunt | Pure Coordination | Clean Up | Prisoners' Dilemma | Rational Coordination | Chicken Game |
|---|:---:|:---:|:---:|:---:|:---:|:---:|
| Temporal Coordination | ✗ | ✗ | ✓ | ✗ | ✗ | ✗ |
| Reciprocity | ✓ | ✓ | ✓ | ✓ | ✗ | ✓ |
| Deception | ✓ | ✗ | ✓ | ✓ | ✗ | ✓ |
| Fair Resource Sharing | ✗ | ✗ | ✓ | ✗ | ✗ | ✗ |
| Convention Following | ✓ | ✓ | ✓ | ✗ | ✓ | ✓ |
| Task Partitioning | ✗ | ✗ | ✓ | ✓ | ✗ | ✗ |
| Trust & Partnership | ✓ | ✗ | ✗ | ✗ | ✗ | ✓ |
| Free Riding | ✗ | ✗ | ✓ | ✗ | ✗ | ✗ |

free riding as depicted in Table 1. An agent performing well in such environments indicates that its behaviors exhibit these properties. In Fig. 4, the agent's observation is shown in the green box to the lower left of the state (*i.e.*, the whole image). The agent is in the lower middle of the observation. The deep neural network architecture of the agent's policy is shown on the left. More information about substrates, scenarios, neural network architectures and training details can be found in Appx. D.

**Baselines.** Our baselines are MAPPO (Yu et al., 2021), MAA2C (Papoudakis et al., 2021), OPRE (Vezhnevets et al., 2020), heuristic fictitious self-play (HFSP) (Heinrich, 2017; Berner et al., 2019) and RandNet (Lee et al., 2019). MAPPO and MAA2C are MARL methods that achieved outstanding performance in various multi-agent scenarios (Papoudakis et al., 2021). OPRE was proposed for the generalization of MARL. RandNet is a general method for the generalization of RL by introducing a novel component in the convolutional neural network. HFSP is a general self-play method for obtaining equilibria in competitive games, we use it by using the policies saved by RPM.

**Training setup.** We use 6 representative substrates (Fig. 5) to train MARL policies and choose some evaluation scenarios from each substrate as our evaluation testbed. The properties of the environments are listed in Table 1. We train agents in Melting Pot substrates for 200 million frames with 3 random seeds for all methods. Our training framework is distributed with 30 CPU actors to collect experiences and 1 GPU for the learner to learn policies. We implement our actors with Ray (Moritz et al., 2018) and the learner with EPyMARL (Papoudakis et al., 2021). We use mean-std to measure the performance of all methods. The bold lines in all figures are mean values, and the shades stand for the standard deviation. Due to a limited computation budget, it is redundant to compare our method with other methods, such as QMIX (Rashid et al., 2018) and MADDPG (Lowe et al., 2017) as MAPPO outperforms them. All experiments are conducted on NVIDIA A100 GPUs.

## 6.2 EXPERIMENT RESULTS

To answer **Q1**, we present the evaluation results of 17 Melting Pot evaluation scenarios in Fig. 6. Our method can boost MARL in various evaluation scenarios, which have different properties, as shown in Table 1. In Chicken Game (CG) 1-2 (the number stands for the number of the evaluation scenario of Chicken Game), RPM outperforms its counterparts by a convincing margin. HFSP performs no better than RPM. RandNet gets around 15 evaluation mean returns on Chicken Game (CG) 1. MAA2C and OPRE perform nearly random (the red dash lines indicate the random result) in the two scenarios. In Pure Coordination (PC) 1-3, Rational Coordination (PC) 1-3 and Prisoners' Dilemma (PD) 1-3, most baselines perform poorly. In Stag Hunt (SH) 1-3 and Clean Up (CU) 1-2, MAPPO and MAA2C perform unsatisfactorily. We can also find that HFSP even gets competitive performance in Stag Hunt (SH) 1-3. However, HFSP performs poorly in Pure Coordination (PC) 1-3, Rational Coordination (RC) 1-3 and Prisoners' Dilemma (PD) 1-3. Therefore, the vanilla self-play method cannot directly be applied to improve the generalization of MARL methods. In summary, RPM boosts the performance up to around 818% on average compared with MAPPO on 6 evaluation scenarios. To answer **Q2**, we present experimental results of the impact of $\psi$ and the sampling ratio in HFSP in the following.

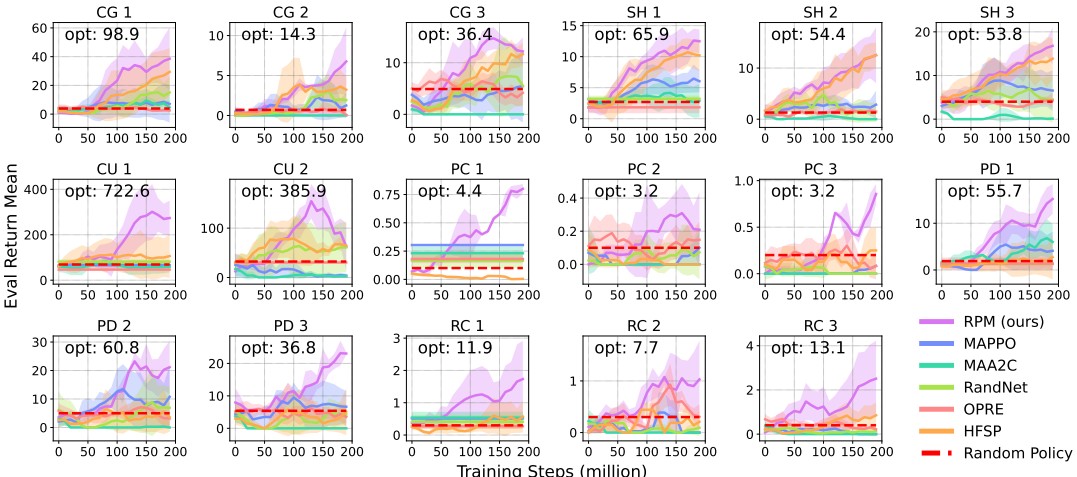

Figure 6: Evaluation results of RPM and baselines in 17 scenarios. The red dash horizontal lines indicate the results of random policy. The optimal (opt) values are shown in each sub-figure and were gathered from (Leibo et al., 2021), which an exploiter generated. The exploiter was trained in the evaluation scenarios with RL methods, and the training time steps were 1,000 M.

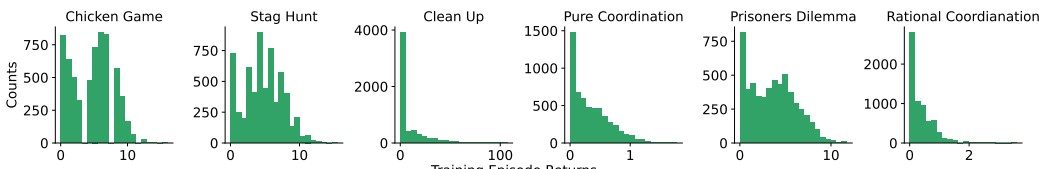

Figure 7: Histograms of training episode returns.

## 6.3 ABLATION STUDY

**The Impact of $\psi$.** To investigate which value of $\psi$ has the greatest impact on RPM performance, we conduct ablation studies by (i) removing ranks and sampling from the checkpoint directly; (ii) reducing the number of ranks by changing the value of $\psi$. As shown in Fig. 8, without ranks (sampling policies without ranks randomly), RPM cannot attain stable performance in some evaluation scenarios. Especially in Pure Coordination (PC) 1-3, the result is low and has a large variance. In RPM, choosing the right interval $\psi$ can improve the performance, as shown in the results of Pure Coordination (PC) 1-3 and Prisoners' Dilemma (PD) 1-3, showing that the value of $\psi$ is important for RPM. We summarize the results and values of $\psi$ in Table 2 and Table 3.

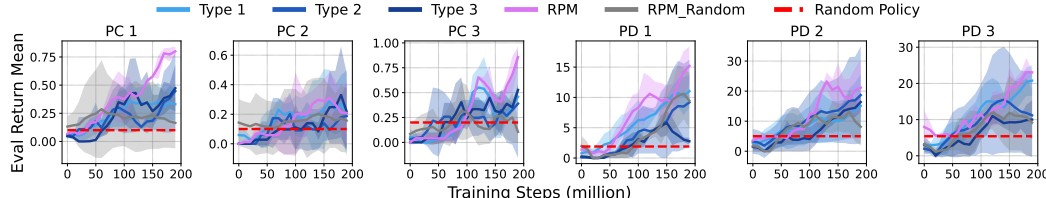

Figure 8: Ablation Studies: the performance of RPM with 3 types of $\psi$ and Random sampling (without ranks).

Table 2: Ablation study: the averaged value of the last three evaluation episode returns. Curves are in Fig. 8.

| Eval Scenarios | RPM | Random | Types of $\psi$ | | |
|---|---|---|---|---|---|
| | | | 1 | 2 | 3 |
| Pure Coordination 1 | 0.78 | 0.18 | 0.33 | 0.39 | 0.42 |
| Pure Coordination 2 | 0.23 | 0.16 | 0.24 | 0.17 | 0.27 |
| Pure Coordination 3 | 0.70 | 0.19 | 0.37 | 0.33 | 0.42 |
| Prisoners' Dilemma 1 | 13.90 | 10.11 | 10.70 | 8.70 | 3.20 |
| Prisoners' Dilemma 2 | 19.60 | 10.41 | 13.76 | 17.74 | 14.96 |
| Prisoners' Dilemma 3 | 22.31 | 10.28 | 19.80 | 11.74 | 9.76 |

Table 3: $\psi$ values. $\psi^*$ indicates the values of $\psi$ used to get results in Fig. 6.

| Eval Scenarios | $\psi^*$ | Types of $\psi$ | | |
|---|---|---|---|---|
| | | 1 | 2 | 3 |
| Pure Coordination 1 | 0.01 | 0.1 | 0.5 | 1 |
| Pure Coordination 2 | 0.01 | 0.1 | 0.5 | 1 |
| Pure Coordination 3 | 0.01 | 0.1 | 0.5 | 1 |
| Prisoners' Dilemma 1 | 0.02 | 0.2 | 1 | 5 |
| Prisoners' Dilemma 2 | 0.02 | 0.2 | 1 | 5 |
| Prisoners' Dilemma 3 | 0.02 | 0.2 | 1 | 5 |

**The Sampling Ratio in HFSP** HFSP shows comparable results in some scenarios in Figure 6. In Figure 6, the sampling ratio of HFSP is 0.3. We are interested in studying the impact of the sampling

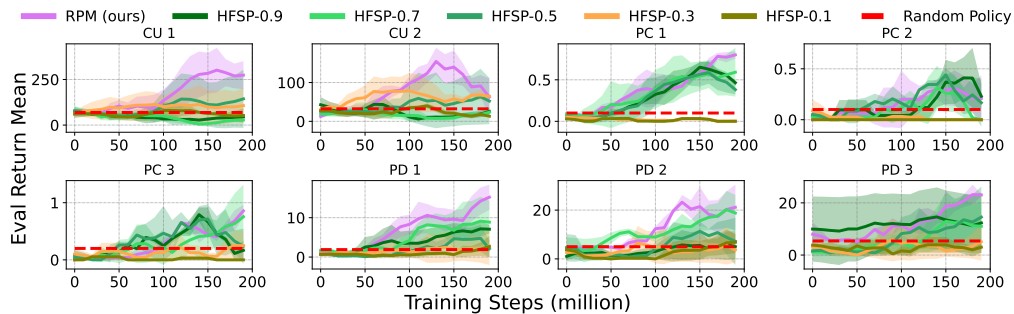

Figure 9: Ablation Studies: the results of HFSP with different sampling ratios.

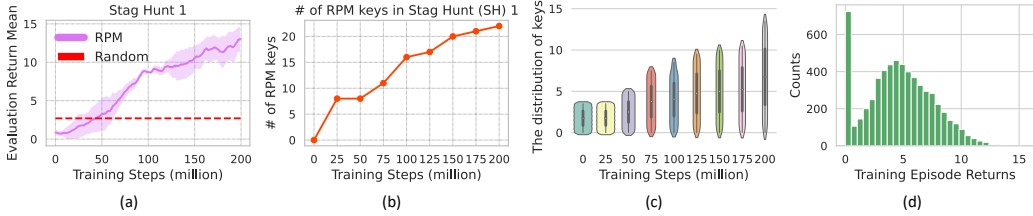

Figure 10: Results analysis. **(a)** The evaluation results of RPM on Stag Hunt (SH) 1; **(b)** The number of RPM keys during training; **(c)** The distribution of the keys of RPM during training; **(d)** The histogram of the keys of RPM at timestep 200M during training.

ratio in HFSP on evaluation performance. We conduct experiments in CU 1 and 2, PC 1 and 3 and PD 1 and 3. The sampling ratio list is $[0.9, 0.7, 0.5, 0.3, 0.1]$. We use the default training setup and use 3 random seeds. HFSP shows comparable results in PC 2 and 3, but its performances are poor in CU 1 and 2 and PD 2 and 3. As shown in Figure 9, HFSP heavily relies on the sampling ratio. HFSP should be carefully tuned on each substrate to attain good performance, which is not feasible. In contrast, RPM is stable (the sampling ratio is 0.5) on all substrates. HFSP can also perform well in substrates such as PC and PD, where the return-checkpoint count distribution is more uniform. The absence of ranks leads to the frequent sampling of policies with high count values in substrates that have skewed return-checkpoint count distribution, thereby reducing the diversity of training data. Such distributions typically comprise a large number of policies with suboptimal performance.

## 6.4 CASE STUDY

We showcase how RPM helps to train the focal agents to choose the right behaviors in the evaluation scenario after training in the substrate. To illustrate the trained performance of RPM agents, we use the RPM agent trained on Stag Hunt and run the evaluation on Stag Hunt 1. In Stag Hunt, there are 8 agents. Each agent collects resources that represent 'hare' (red) or 'stag' (green) and compares inventories in an interaction, *i.e.*, encounter. The results of solving the encounter are the same as the classic Stag Hunt matrix game. In this environment, agents are facing tension between the reward for the team and the risk for the individual. In Stag Hunt 1, One focal agent interacts with seven pretrained background agents. All background agents were trained to play the 'stag' strategy during the interaction[1]. The optimal policy for the focal agent is also to play 'stag'. However, it is challenging for agents to detect other agents' strategy since such a behavior may not persist in the substrate. Luckily, RPM enables focal agents to behave correctly in this scenario.

To answer **Q3**, we present the analysis of RPM on the substrate Stag Hunt and its evaluation scenario SH 1 in Fig. 10. We can find that in Fig. 10 (b), the number of the keys in RPM is growing monotonically during training and the maximum number of the keys in RPM is over 20, showing that agents trained with RPM discover many novel patterns of multi-agent interaction and new keys are created and the trained models are saved in RPM. Meanwhile, the evaluation performance is also increasing in SH 1 as depicted in Fig. 10 (a). In Fig. 10 (c), it is interesting to see that the distribution of the keys of RPM is expanding during training. In the last 25 million training steps, the last distribution of RPM keys covers all policies of different performance levels, ranging from 0 to 14. By utilizing RPM, agents can collect diversified multi-agent trajectories for multi-agent training. Fig. 10 (d) demonstrates the final histogram of RPM keys after training. There are over 600 trained policies that have a small value of keys. Since agents should explore the environment at the early

---

[1]This preference was trained with pseudo rewards by Leibo et al. (2021) and the trained models are available at this link: `https://github.com/deepmind/meltingpot`

stage of training, it is reasonable to find that many trained policies of RPM keys have low training episode returns. After 50 million training steps, RPM has more policies with higher training episode returns. Note that the maximum training episode return of RPM keys is over 14 while the maximum mean evaluation return of RPM shown in Fig. 10 (a) is around 14.

Our experiments show that training policies with good performance in the substrate is crucial for improving generalization performance in the evaluation scenarios. When MARL agents perform poorly in the substrate, the evaluation performance will also be inferior or random, making it hard to have diversified policies. We show the results in Appx. E.

## 7   RELATED WORKS

Recent advances in MARL (Yang & Wang, 2020; Zhang et al., 2021) have demonstrated its success in various complex multi-agent domains, including multi-agent coordination (Lowe et al., 2017; Rashid et al., 2018; Wang et al., 2021b), real-time strategy (RTS) games (Jaderberg et al., 2019; Berner et al., 2019; Vinyals et al., 2019), social dilemma (Leibo et al., 2017; Wang et al., 2018; Jaques et al., 2019; Vezhnevets et al., 2020), multi-agent communication (Foerster et al., 2016; Yuan et al., 2022), asynchronous multi-agent learning (Amato et al., 2019; Qiu et al., 2022), open-ended environment (Stooke et al., 2021), autonomous systems (Hüttenrauch et al., 2017; Peng et al., 2021) and game theory equilibrium solving (Lanctot et al., 2017; Perolat et al., 2022). Despite strides made in MARL, training generalizable behaviors in MARL is yet to be investigated.

Recently, generalization in RL (Packer et al., 2018; Song et al., 2019; Ghosh et al., 2021; Lyle et al., 2022) has achieved much progress in domain adaptation (Higgins et al., 2017) and procedurally generated environments (Lee et al., 2019; Igl et al., 2020; Zha et al., 2020). However, there are few works of generalization in MARL domains (Carion et al., 2019; Vezhnevets et al., 2020; Mahajan et al., 2022; McKee et al., 2022). Recently, Vezhnevets et al. (2020) propose a hierarchical MARL method for agents to play against opponents it hasn't seen during training. However, the evaluation scenarios are only limited to simple competitive scenarios. Mahajan et al. (2022) studied the generalization in MARL empirically and proposed theoretical findings based on successor features (Dayan, 1993). However, no method to achieve generalization in MARL was proposed in (Mahajan et al., 2022).

Ad-hoc team building (Stone & Kraus, 2010; Gu et al., 2021) models the multi-agent problem as a single-agent learning task. In ad-hoc team building, one ad-hoc agent is trained by interacting with agents that have fixed pretrained policies and the non-stationarity issue is not severe. However, in our formulation, non-stationarity is the main obstacle to MARL training. In addition, there is only one ad-hoc agent evaluated by interacting agents that are unseen during training, while there can be more than one focal agent in our formulation as defined in Definition 2, thus making our formulation general and challenging. There has been a growing interest in applying self-play to solve complex games (Heinrich et al., 2015; Silver et al., 2018; Hernandez et al., 2019; Baker et al., 2019); however, its value in enhancing the generalization of MARL agents has yet to be examined. Due to space constraints, we discuss meta-learning (Al-Shedivat et al., 2018; Kim et al., 2021) and population-based training (Strouse et al., 2021; Lupu et al., 2021; Tang et al., 2021) works in Appx. F.

## 8   CONCLUSION, LIMITATIONS AND FUTURE WORK

In this paper, we consider the problem of achieving generalizable behaviors in MARL. We first model the problem with Markov Game. To train agents that can interact with agents that possess unseen policies. We propose a simple yet effective method, RPM, to collect diversified multi-agent interaction data. We save policies in RPM by ranking the training episode return. Empirically, RPM significantly boosts the performance of MARL agents in various Melting Pot evaluation scenarios.

RPM's performance is dependent on the appropriate value of $\psi$. Several attempts may be needed to determine the correct value of $\psi$ for RPM. We are interested in discovering broader measures for ranking policies that do not explicitly consider the training episode return. Recently, there has been a growing interest in planning in RL, especially with model-based RL. We are interested in exploring the direction of applying planning and opponent/teammate modelling for attaining generalized MARL policies for future work. Agents are engaged in complex interactions in multi-agent scenarios. Devising novel self-play methods is our future direction.

ETHICS STATEMENT

We addressed the relevant aspects in our conclusion and have no conflicts of interest to declare.

REPRODUCIBILITY STATEMENT

We provide detailed descriptions of our experiments in the appendix and list all relevant parameters in Table 4 and Table 5 in Appx. D. The code can be found at this link: `https://sites.google.com/view/rpm-iclr2023/`.

ACKNOWLEDGMENTS

We would like to thank the anonymous reviewers for their suggestions. We thank the support from Xinyi Wan, Jiahao Ji and Xiangfan Li of the infrastructure team at Sea AI Lab. Wei Qiu and Bo An are supported by the National Research Foundation, Singapore under its Industry Alignment Fund – Pre-positioning (IAF-PP) Funding Initiative. Any opinions, findings and conclusions or recommendations expressed in this material are those of the author(s) and do not reflect the views of National Research Foundation, Singapore.

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
