# OpenReview forum: "RPM: Generalizable Multi-Agent Policies for Multi-Agent Reinforcement Learning"
_ICLR.cc/2023/Conference — ICLR 2023 poster_

### Official Review · Reviewer_7r99 · 2022-10-21

**Confidence:** 3
**Correctness:** 3
**Technical Novelty And Significance:** 2
**Empirical Novelty And Significance:** 3
**Recommendation:** 5

**Clarity, Quality, Novelty And Reproducibility:**

The clarity of this paper is good, since the whole paper is well written.

The novelty of this paper may not be adequate, since it is just an intuitive extension from the prior self-play framework with no reasonable explanations.

The originality of the paper is moderate as far as I know. However, I am not sure whether the similar idea has appeared in the past (since it is so direct and intuitive)?

The reproducibility of this paper is good, since it provides both codes and experimental setups.

Overall, the quality of this paper is not bad.

**Strength And Weaknesses:**

## Strength
1. This paper is generally well written and the description of the proposed method is clear.
2. The motviation of this paper is clear and I believe this is a meaningful work.
3. The proposed method is demonstrated to improve the performance of generalization.
4. The ablation study well spots the limitation of the proposed method, so that it is transparent and easy for the followers to improve it.
5. The comparisons to the prior works are sufficient.

## Weaknesses
1. The proposed method is so heuristic (though it cannot be a reason to reject it), so that it is unclear why it works.
2. Although the authors claim that the diversified multi-agent trajectories can resemble trajectories generated by the interaction with agents possessing unknown policies in the evaluation scenario, this explanation is unconvincing to me. The most critical reason is that the background agent is pre-trained, so it is possible that the policy space of trained agents **completely** deviates from the policy space of pre-trained agents (with no intersection). The authors should give a better explanation.
3. The proposed RPM is highly dependent on the choice of a hyperparameter $\psi$ as shown in ablation study. Could the authors give a brief proposal to address this issue?
4. As shwon in Fig. (e) in ablation study, the mode of the final distribution of keys of RPM is almost the mean of returns. It can be estimated that the effect of RPM could make the MARL policies adapt the average performance. This might lead to the issue as the authors discussed in the paper that MARL agents' performances will affect the evaluation performance.

**Summary Of The Paper:**

This paper proposes a simple method called ranked policy memory (RPM) that can be plugged in any existing MARL algorithms to solve the generalization problem of MARL. The main idea of RPM is maintaining a look-up memory of the history rollout policies which are ranked by the training episode return. At each episode, the rollout policies are uniformly sampled from RPM. The objective of this process is to keep the diversity of agents' policies and resemble the the unknown policies that may appear in the evaluation scenarios. The performance of RPM is empirically verified by experiments and abnalation studies.

**Summary Of The Review:**

The proposed method in this paper performs better than the baselines and it is demonstrated in the ablation study that it should be effective. Nevertheless, the principle of the effectiveness is blurry.

---

> ### Author Response · Authors · 2022-11-13
> **Response to Reviewer 7r99**
>
> Dear reviewer, we thank you for providing precious comments. We summarize the questions and present our responses below:
>
> ---
>
> **Q1: Why the method can work?**
>
> **A1:** We believe the main reasons are: (i) To improve generalization in MARL, diversified agents’ behaviors are needed for MARL training; (ii) there are many checkpoints in self-play. Sampling n policies from the big checkpoint pool cannot attain a group of agents with diversified policies compared with RPM.
>
> ---
>
> **Q2: Why the diversified multi-agent trajectories can resemble trajectories generated by the interaction with agents possessing unknown policies in the evaluation scenario?**
>
> **A2:** Since agents cannot use the multi-agent interaction data generated in the evaluation scenario to train the MARL policies, it is of great significance to generate diversified data for MARL training. A similar learning paradigm can be also found in ML and computer vision, where data augmentation is needed to improve the trained predictor’s generalization under unseen data.
>
> ---
>
> **Q3: The proposed RPM is highly dependent on the choice of a hyperparameter**
>
> **A3:** We discussed it as a limitation of our method in Section 7: Conclusion, Limitations and Future Work. As RPM is a new method, we need 1-3 attempts to get the appropriate $\psi$, which is similar to tuning the learning in deep learning. We also have a rule of thumb to reduce the attempts. We use the minimal non-zero reward the agent can receive as the $\psi$. We can get such a reward from the setting of the substrate. Then we can compare the performance of each $\psi$ by scaling it up to the power of 10 (or 5) or reducing it to the power of 1/10 (or 1/5). It would not take many attempts compared with exhausting all the $\psi$  values.

---

> > ### Comment · Reviewer_7r99 · 2022-11-18
> > **Thank you for your reply**
> >
> > Dear Authors,
> >
> > I have carefully read the response, though most of them are the heuristic explanations which are not the ones I would like to see. I will discuss the paper with other reviewers later to decide whether I raise my score. At the moment, I can only keep my original decision.

---

> ### Author Response · Authors · 2022-11-17
> **Dear Reviewer 7r99, did our response answer your questions?**
>
> Dear Reviewer 7r99,
>
> We thank you again for your comments on our paper. Did our response answer your questions and address the concerns? As the response system will be closed soon within ONE day. If you have no more questions, we would appreciate it if you could kindly consider raising the score. More questions and discussions on our paper are always welcome!
>
> Sincerely yours,
>
> Authors

---

### Official Review · Reviewer_3kor · 2022-10-23

**Confidence:** 4
**Correctness:** 2
**Technical Novelty And Significance:** 3
**Empirical Novelty And Significance:** 3
**Recommendation:** 6

**Clarity, Quality, Novelty And Reproducibility:**

The paper is in general well written and particularly easy to follow.

There is a small type saying $\phi$ is an integer (two lines above equation (2)). However, according to Table 2, this isn't the case.

The reproducibility is good and the website is well-prepared.

Regarding the novelty, I think the paper can be much stronger if the authors could provide an in-depth analysis of the necessity of the discretized categorization of policy skill level.



**Strength And Weaknesses:**

## Strength

The paper is clearly written and easy to follow. Although the overall idea isn't groundbreaking, it is still novel to store policies in a discretized memory and perform training over randomly sampled policies. This is algorithmically different from the classical way of using policy archives in MARL, which typically follows the framework of fictitious self-play. In addition, the selected testbed is sufficiently challenging. The proposed method could be well served as a baseline for the following works.

## Weakness

### Missing citations

In the related work section, the authors claim that "_there is no method proposed to achieve generalization in MARL_", which, to the best of my knowledge, is not true. It is true that there isn't a systematic study on the setting of general sum and more than two agents, but there are indeed a lot of works on relatively more narrow domains. For example, [1] adopts the same idea of using past policies to generate diverse interactions so that the learned policy can generalize to humans. The difference is that [1] conducts two stages: i.e.,  first, create a policy population of different skill levels and then train an adaptive policy from scratch to compete with diverse partners so that it can generalize during evaluation. [2] also considers the stag hunt game and follows a similar training paradigm to [1] to train a policy that can adapt to cooperative or non-cooperative partners. [3] adopts a similar population-based training framework to this work but improves the generalization ability of policies by promoting policy diversity. I think the paper should also carefully discuss these related works in the paper.

[1] Collaborating with Humans without Human Data, DJ Strouse, Kevin R. McKee, Matt Botvinick, Edward Hughes, Richard Everett, NeurIPS 2021.

[2] Discovering Diverse Multi-Agent Strategic Behavior via Reward Randomization, Zhenggang Tang, Chao Yu, Boyuan Chen, Huazhe Xu, Xiaolong Wang, Fei Fang, Simon Shaolei Du, Yu Wang, Yi Wu, ICLR 2021

[3] Trajectory Diversity for Zero-Shot Coordination, Andrei Lupu, Brandon Cui, Hengyuan Hu, Jakob Foerster, ICML 2021

### Relationship to Fictitious Self-Play (FSP)
I have been carrying by the same question throughout the reading of this paper. Although I do see experiments between the proposed method and FSP, I could still hardly understand **why** (at least intuitively) FSP is worse than RPM. I think the paper can be much improved if this question can be carefully answered and discussed. Some of my thoughts on this question are listed below.

1. **The performance of FSP should be tuned**. In Fig 6, the performance of HSP is substantially worse than MAPPO and it is even worse than _random_. I checked the appendix, which states that HSP uses a surprisingly high past sampling rate of 30%. Particularly in the case of multiple agents, such a hyper-parameter choice could largely hurt MARL training. As a reference, [4] uses a 5% past sampling rate. I do think this hyper-parameter should be carefully tuned to ensure a fair comparison, considering the fact that FSP is perhaps the most important baseline to compare with. In particular, the RPM-random baseline achieves comparable performance to RPM in the prisoner's dilemma game while FSP is even worse than random.


[4] Emergent Tool Use From Multi-Agent Autocurricula, Bowen Baker, Ingmar Kanitscheider, Todor Markov, Yi Wu, Glenn Powell, Bob McGrew, Igor Mordatch, ICLR 2020

2. **The motivation for using a discretized category (rank)**. I'm definitely convinced that we should run MARL training with partners with different skill levels. And, it is great to see in the ablation studies that RPM-random baseline works worse. However, I don't think the paper at any place **explains** and **motivates** why the use of a discretized rank is _necessary_. Intuitively, why wouldn't past sampling achieve the same effect? Note that RPM-random is indeed comparable to RPM in the prisoner's dilemma case. By contrast, why does RPM-random works poorly in pure cooperation? Note that random sampling achieves strong performances in the overcooked game [1], which is also a purely cooperative game. I do think some in-depth analysis should be conducted.
A possible reason that I can imagine is as follows. Based on the histogram in fig 8(e), the distribution of policy skill levels is not uniform. Those high-reward policies are much rarer than sub-optimal ones. So a naive random sampling may hardly choose those high-reward policies, which accordingly makes training slow. This could be a fair argument. However, if we exclude those policies with the highest rewards (e.g., return > 10), those sub-optimal policies are distributed pretty much uniformly in fig.8(e) to some extent.  So, couldn't this issue (uneven distribution of skill levels) be just solved by FSP with a properly tuned past sampling rate? With a properly tuned rate, the probability of choosing a recent high-reward policy and choosing a poor past policy can be well balanced, if fig 8(e) is a generic diagram for most MARL applications. So, in addition to tuning the FSP baseline better, I would suggest the authors study the policy distribution over every scenario to have a better **understanding** of why random sampling would fail and why a discretized category is necessary.

3. **A fair comparison**. This is a final and possibly repetitive comment. RPM requires careful tuning over the value of $\phi$, and RPM would simply generate RPM-random if $\phi$ approaches 0. Unfortunately, there isn't a generic way to choose a good $\phi$ (at least not in the current draft), so I would believe a similar amount of tuning efforts should be made for FSP on past sampling rates for a fair comparison.


**Summary Of The Paper:**

This paper considers improving the generalization capability of MARL agents. The idea is to maintain an archive of policies based on their discretized returns. In each episode, agents from random return categories are selected to perform RL training, which ensures that each episode can contain diversified agent interactions. Experiments are conducted on the melting pot environment against a collection of baselines.

Although the idea of using policies of different skill levels for improved generalization isn't new in MARL, the execution of this idea in the setting of general multi-agent games is neat and intuitive.


+++++++++++++++++++++++ post rebuttal +++++++++++++++++++++++
The authors have included additional results on the baselines, which makes the paper more compete. Therefore, I decided to update my score from 5 to 6.


**Summary Of The Review:**

This is a well-written paper with a neat idea. The current content is very close to the bar of acceptance in my opinion and can be further improved if more in-depth analysis can be provided.

---

> ### Author Response · Authors · 2022-11-13
> **Response to Reviewer 3kor 1/2**
>
> Dear reviewer, we thank you for your insightful comments and constructive feedback. We summarize your questions and provide our responses below.
>
> ---
>
> **Q1: In the related work section, the authors claim that "there is no method proposed to achieve generalization in MARL", which, to the best of my knowledge, is not true.**
>
> **A1:** We did not mean to claim that “there is no method proposed to achieve generalization in MARL”. We did not anticipate that it would cause misunderstanding. That sentence discussed the drawback of [1]. We updated our paper by changing that sentence to “However, no technique to achieve generalization in MARL has been proposed in (Mahajan et al., 2022)”.
>
> We also noticed the papers you mentioned in the comment [2, 3, 4]. We discussed them in Appendix F in our revised paper. Here, we present our discussion. Fictitious Co-Play (FCP) proposed in [2] aims to learn policies for a two-agent cooperative game. It is a **two-stage** method. In the first stage, $n$ agents are trained independently with different random seeds via self-play. It needs n seeds and much more extra computation. In the second stage, an FCP agent is trained by interacting with the trained policies of $n$ agents. Another extra run of training is also needed. However, our method, RPM, is an end-to-end training method for social dilemmas, competitive, cooperative and even mixed environments with more than two agents. It needs only one run training (i.e., end-to-end one step) by utilizing fictitious self-play to sample n policies for each agent without maintaining n populations of agents.
>
> The paper [3] considered the problem of zero-shot coordination and proposed a method to achieve diversity via population-based training (PBT) method. In contrast, our work aims to achieve the generalization of coordination, competition and social dilemmas in multi-agent systems via our novel ranked policy memory method. The proposed method in [3] cannot be applied to competition and social dilemma scenarios. Besides that, PBT needs much more computation, which could be computationally expensive for large-scale multi-agent scenarios. The paper [3] is also pointed out that “self-play (SP) agents control their own trajectory distribution during training, each policy typically only performs well on this exact distribution.” We believe that is the key issue of the self-play method. The issue can be alleviated by introducing ranks, and each agent loads its previous policy for multi-agent self-play.
>
> The RPG method proposed in [4] is a population-based training method without self-play. It is highly dependent on the randomized reward function. For simple grid world scenarios in [4], conducting randomized reward function perturbations is not challenging. However, it is non-trivial to find proper reward function perturbations for complex scenarios.
>
> ---
>
> **Q2: Presenting results of HFSP with 5% past sampling rate.**
>
> **A2:** We use the probability of 0.05 sampling rate to sample past policies and 0.95 to sample policies from the latest top-n ranks. The results are even not as good as the previous results of HFSP (the probability of sampling past policies and sample the latest top-n ranks are 0.3 and 0.7, respectively).
>
> ---
>
> **Q3: The motivation for using a discretized category (rank).**
>
> **A3:** Our motivations for using a discretized category are: (i) there are many checkpoints in self-play. Sampling $n$ policies from the big checkpoint pool cannot attain a group of agents with diversified policies compared with RPM; (ii) To improve generalization in MARL, diversified agents’ behaviors are needed for MARL training.
>
> We agree that RPM-random is not better than RPM because the distribution of policy skill levels is not uniform. We noticed that if we exclude those policies with the highest rewards (e.g., return > 10) in Fig. 8, the remaining distribution is actually not an uniform distribution because the counts of each rank is not identical. It is also worth noting that the returns are the episode return during training. It is not the evaluation episode return.
>
> The final RPM distributions for each substrate are also shown in the following anonymous link: https://sites.google.com/view/rpm-iclr2023/home. Finding that RPM keys have long tail distributions is interesting. As a result, if we directly sample policies from all the previous policies without ranks, we can easily sample a large number of policies with poor performance, which will not contribute to the diversification of the data on multi-agent interactions. As an alternative, we can first sample keys, then sample policies for each key, which can solve the problem mentioned earlier.
>
> Moreover, in the figure, the distribution of RPM keys in prisoner’s dilemma is more like a uniform distribution compared other distributions if we exclude the count of key=0. It can explain why RPM-random, despite being less effective RPM, performs well in the prisoner's dilemma.

---

> > ### Author Response · Authors · 2022-11-13
> > **Response to Reviewer 3kor 2/2**
> >
> > ---
> >
> > **Q4: There is a small type saying ϕ is an integer (two lines above equation (2)).**
> >
> > **A4:** We have fixed the typo in the revised paper. It can be either integer or float values.
> >
> > ---
> >
> > **Reference:**
> >
> > [1] Mahajan, A., Samvelyan, M., Gupta, T., Ellis, B., Sun, M., Rocktäschel, T., & Whiteson, S. (2022). Generalization in Cooperative Multi-Agent Systems. arXiv preprint arXiv:2202.00104.
> >
> > [2] Collaborating with Humans without Human Data, DJ Strouse, Kevin R. McKee, Matt Botvinick, Edward Hughes, Richard Everett, NeurIPS 2021.
> >
> > [3] Trajectory Diversity for Zero-Shot Coordination, Andrei Lupu, Brandon Cui, Hengyuan Hu, Jakob Foerster, ICML 2021
> >
> > [4] Discovering Diverse Multi-Agent Strategic Behavior via Reward Randomization, Zhenggang Tang, Chao Yu, Boyuan Chen, Huazhe Xu, Xiaolong Wang, Fei Fang, Simon Shaolei Du, Yu Wang, Yi Wu, ICLR 2021

---

> > > ### Comment · Reviewer_3kor · 2022-11-16
> > > **Thanks for the response**
> > >
> > > I'm really excited to see the additional histogram results. They are really interesting. I would strongly encourage the authors to include some discussions on this observation in the main paper to make the paper stronger.
> > >
> > > Regarding the HFSP experiments, actually, I'm not suggesting that 0.05 past sample rate is the best universal optimal choice. I was trying to say that some tuning efforts should be also made for HFSP for a fair comparison. For example, possibly some grid search process (e.g., similar to table 3) should be done rather than simply saying we choose 0.3 because 0.3 is better than 0.05. For example, at least for the clean up plot in Figure 6, the 0.3 past sample rate looks clearly sub-optimal.
> > >
> > > Besides these, I think the paper is pretty close to the acceptance threshold.

---

> > > > ### Author Response · Authors · 2022-11-25
> > > > **More HFSP results**
> > > >
> > > > Dear reviewer,
> > > >
> > > > Thank you for the suggestion. We will put that figure and the finding into the final version. Tuning HFSP on Melting Pot requires a significant amount of time and computation resources. We present the results of HFSP with $[0.1, 0.3, 0.5, 0.7, 0.9]$ sampling ratios (3 seeds, see the figure in the project link shown in our previous response). It shows that HFSP heavily relies on the sampling ratio. HFSP should be carefully tuned on each substrate to attain good performance, which is not feasible.
> > > >
> > > > In contrast, RPM is stable (sampling ratio is 0.5) and we did not put extra effort into tuning RPM. HFSP can also perform not bad in substrates such as Pure Coordination and Prisoners Dilemma, where the return-checkpoint count distribution is more uniformly distributed. We also discussed this finding in our previous response.
> > > >
> > > > Please let us know if you have any questions.
> > > >
> > > > Sincerely yours,
> > > >
> > > > Authors.

---

> > > > > ### Comment · Reviewer_3kor · 2022-11-25
> > > > > **Nice update**
> > > > >
> > > > > I do think the paper is now in a much more complete and sound form. The result that HFSP would depend on the ratio is consistent with my expectation. I do think being frank and transparent about this fact is critical
> > > > >
> > > > > I'm happy to raise my score.

---

### Official Review · Reviewer_GwoH · 2022-10-24

**Confidence:** 4
**Correctness:** 3
**Technical Novelty And Significance:** 2
**Empirical Novelty And Significance:** 3
**Recommendation:** 6

**Clarity, Quality, Novelty And Reproducibility:**

**Clarity:** The paper is well-written and conveys the main insights well.
**Quality & Novelty:** I agree that RPM is new regarding the self-play notion. The setting and objective may overlap with multi-agent meta-learning methods.
**Reproducibility:** The source code is provided to reproduce the results.

**Strength And Weaknesses:**

**Strengths:**
1. The paper is generally well-written and conveys the methods clearly. The figures are also helpful in understanding the method.
2. While RPM is a relatively simple algorithm based on the episodic return ranking system, it shows effectiveness in multiple scenarios. As such, the algorithm is directly applicable to other settings/methods.

**Questions:**
1. The problem formation and objective in Section 3 are closely related to meta-learning in MARL (Al-Shevidat et al., ICLR 2018; Kim et al., ICML 2021), where the goal is to train a meta-agent with a population of other agents such that the meta-agent can adapt well when interacting with a new agent at meta-testing. The main difference between this paper and meta-MARL is that the focal agents are not allowed to fine-tune their policies during the evaluation in this paper, while meta-agents are allowed to fine-tune their policies. Because both settings are concerned with generalization, I would like to ask discussion comparing the two settings.
2. Would RPM benefit from using more complicated ranking systems (e.g., TrueSkill)?
3. Adding lines that denote optimal values in Figure 6 (i.e., if ideal generalization is possible) can help identify the gap and the amount of generalization performed by RPM.

**References:**
* Maruan Al-Shedivat, Trapit Bansal, Yuri Burda, Ilya Sutskever, Igor Mordatch, Pieter Abbeel. Continuous Adaptation via Meta-Learning in Nonstationary and Competitive Environments. ICLR 2018
* Dong-Ki Kim, Miao Liu, Matthew Riemer, Chuangchuang Sun, Marwa Abdulhai, Golnaz Habibi, Sebastian Lopez-Cot, Gerald Tesauro, Jonathan P. How. A Policy Gradient Algorithm for Learning to Learn in Multiagent Reinforcement Learning. ICML 2021

**Summary Of The Paper:**

This paper proposes a new self-play training framework called RPM that focuses on the diversity of the multi-agent population. Specifically, RPM builds the population by maintaining various levels of policies via ranking. Then, RPM trains focal agents that behave well with background agents sampled from the population. The extensive evaluations based on the melting pot domain show the generalization of RPM when interacting with unseen agents in the evaluation scenarios.

**Summary Of The Review:**

Overall, I have a positive evaluation of this paper (score of 6), and I will make a final decision on the recommendation after the authors' response.

---

> ### Author Response · Authors · 2022-11-13
> **Response to Reviewer GwoH**
>
> Dear reviewer, we appreciate your high-quality reviews and constructive feedback. We summarize your questions and present our responses below.
>
> ---
>
> **Q1: The problem formation and objective in Section 3 are closely related to meta-learning in MARL. Discussing and comparing the two settings.**
>
> **A1:** We added the discussion in Appendix F in the paper due to the 9-page limit of the main text.
>
> Our formulation aims to improve the generalization of MARL. The generalization in MARL is closely related to the concept of generalization in machine learning (ML). The substrates in our formulation are the “training dataset,” and the evaluation scenarios in our formulation are the “test dataset.” In evaluation scenarios, focal agents should interact with background agents with unseen policies during training to complete the task.
>
> Meta-learning in MARL [1, 2] aims to address the non-stationarity issue in MARL, a well-known issue that has been extensively studied. To address the issue, [1, 2] adopted the learning-to-learn framework. Typically, Al-Shedivat et al. [1] consider the problem of continuous adaptation in non-stationary environments where agents have few-shot interactions, i.e., the agent must learn from only a limited amount of experience that it can collect before its environment changes. The MAML framework was used to address the issue. Kim et al. [2] proposed a novel meta-multiagent policy gradient theorem that directly accounts for the non-stationary policy dynamics inherent to multiagent learning settings. The proposed meta-multiagent policy gradient theorem explicitly models the learning procedure of the other agent (peer learning) mainly in two-agent settings by considering the sequential dependence of the future parameters of other agents on the meta-agent's parameter.
> In contrast, the previous work [1] ignored it. However, it would be non-trivial to train 3+ meta-agents simultaneously in complex scenarios due to the infinite recursion problem associated with the meta-learning framework in [1]. Therefore, the two works consider **two-agent** settings. However, solving the non-stationarity issue in scenarios with more than two agents is more challenging than in two-agent settings. The performance of the two works in these scenarios is yet to be investigated. Furthermore, adapting the framework of meta-learning in MARL to improve the generalization in MARL is non-trivial.
>
> ---
>
> **Q2: Would RPM benefit from using more complicated ranking systems**
>
> **A2:** RPM can benefit from using proper ranking measures. TrueSkill is a rating system among game players for ranking and matching players in **competition games**, such as StarCraft II and DoTa 2. It was not proposed for ranking policies. However, RPM ranks agents’ policies for self-play. Melting Pot environments are not all competition games. **Most of the environments in Melting Pot are not competition games but cooperative, coordination and social dilemma scenarios.**
>
> ---
>
> **Q3: Adding lines that denote optimal values in Figure 6.**
>
> **A3:** We added the lines that denote the optimal values in Figure 6. The optimal values were gathered from Leibo et al. 2021 [3], which were generated by an exploiter. The exploiter was trained in the evaluation scenarios with the RL method. The training time steps is 1,000 M.
>
> ---
>
> **References:**
>
> [1] Maruan Al-Shedivat, Trapit Bansal, Yuri Burda, Ilya Sutskever, Igor Mordatch, Pieter Abbeel. Continuous Adaptation via Meta-Learning in Nonstationary and Competitive Environments. ICLR 2018
>
> [2] Dong-Ki Kim, Miao Liu, Matthew Riemer, Chuangchuang Sun, Marwa Abdulhai, Golnaz Habibi, Sebastian Lopez-Cot, Gerald Tesauro, Jonathan P. How. A Policy Gradient Algorithm for Learning to Learn in Multiagent Reinforcement Learning. ICML 2021
>
> [3] Leibo, J. Z., Dueñez-Guzman, E. A., Vezhnevets, A., Agapiou, J. P., Sunehag, P., Koster, R., ... & Graepel, T. (2021, July). Scalable evaluation of multi-agent reinforcement learning with melting pot. In International Conference on Machine Learning (pp. 6187-6199). PMLR.

---

> > ### Comment · Reviewer_GwoH · 2022-11-17
> > **Response to Rebuttal**
> >
> > I appreciate the authors for updating the paper based on my questions. The response has addressed my questions.

---

> > > ### Author Response · Authors · 2022-11-18
> > > **Dear Reviewer GwoH, Thanks For Your Acknowledgement**
> > >
> > > Dear Reviewer GwoH,
> > >
> > > Thanks for your acknowledgement.
> > >
> > > Sincerely yours,
> > >
> > > Authors.

---

> ### Author Response · Authors · 2022-11-17
> **Dear Reviewer GwoH, do you have any questions?**
>
> Dear Reviewer GwoH,
>
> We thank you again for recognising the contribution of our work to the MARL. We appreciate your time and hard work in providing valuable comments for our paper. As the response system will be closed soon within ONE day. Please let us know if you have any questions. More questions and discussions on our work are always welcome!
>
> Sincerely yours,
>
> Authors.

---

### Official Review · Reviewer_8Vwu · 2022-10-25

**Confidence:** 5
**Correctness:** 4
**Technical Novelty And Significance:** 3
**Empirical Novelty And Significance:** 3
**Recommendation:** 5

**Clarity, Quality, Novelty And Reproducibility:**

### Clarity
* It is unclear how agents are chosen at evaluation time to be entered into the substrate.
* The focal agent's return is clearly dependent on the co-player trained with, how do you handle this instability during training?

### Quality
* Typos in "newly collected trajecotries and πθb"
* In the ablations diagrams you refer to  \phi type III but do not explain what this is.

### Novelty
Method is sufficiently novel - however it is very similar to [Fictitious Co-Play](https://arxiv.org/abs/2110.08176), the agents stored in the buffer are very similar to a population of agents and thus explaining the differences in related work could be useful.

### Reproducibility
No code is provided, nor is method clear enough for reproducibility.

**Strength And Weaknesses:**

### Strengths

* Idea is simple and good first step in the melting pot game
* Good ablations


### Weaknesses
* No explanation of the RPM update rule.
* Method makes a major assumption that return is enough to distinguish behaviour. A toy example of where this fails would be that grim trigger and tit-for-tat would get similar returns in IPD against a defective agent but intuitively have some variance in behaviour (e.g.  tit-for-tat is forgiving but grim trigger is not)
* Unclear how large a RPM buffer should be
* This method is computationally much more intensive than baselines, some analysis to count the different number of timesteps used in training or when convergence is achieved would also be reasonable.


**Summary Of The Paper:**

The authors propose a method for containing multiple distinct behaviours (or policies) in a single agent. To do this they store a dictionary of policies observed during training, where the key relates to returns observed during training (this is discretised to keep the buffer size reasonable).

This is actually very similar to simply holding a population of agents during training. At the start of training the dictionary (size N)  is initialised with N random policies which are all selected randomly and trained with. RPM uses self-play from its buffer to train more agents. Unlike self-play or population-play their is no risk of degeneracy as values are only overwritten for agents which produce the same return.

**Summary Of The Review:**

Simple method but paper lacks clarity.

---

> ### Author Response · Authors · 2022-11-13
> **Response to Reviewer 8Vwu 1/2**
>
> Dear reviewer, we thank you for your valuable comments on our paper. We summarize your questions and present our responses below.
>
> ---
>
> **Q1: No explanation of the RPM update rule.**
>
> **A1:** It is introduced in the RPM Building paragraph in Sec. 4.1, page 4. We replace RPM Building with RPM Building & Updating to highlight it.
>
> ---
>
> **Q2: Method makes a major assumption that return is enough to distinguish behaviour.**
>
> **A2:** We do not claim that “return is enough to distinguish behavior” in our paper. To improve the generalization performance of MARL, we adopt the self-play framework to gather diversified multi-agent interaction data, i.e., trajectories. We believe that the diversified agent-agent interaction data are generated by agents with different ranks of policies. Naturally, the episode return can be a good measure to rank agents’ behavior. In Strouse et al., 2021 [1], multiple checkpoints of each self-play partner throughout training allowed the pool to represent different “skill”/behavior levels.
>
> ---
>
> **Q3: Unclear how large a RPM buffer should be.**
>
> **A3:** For most of the substrates, there are 20-400 keys and approximately ~6,667 checkpoints in total. We list the number of keys below:
>
>
> | Substrate | # keys  |
> |:--------:|:---:|
> |Chicken Game | 20  |
> |Clean Up         | 80   |
> |Prisoners' Dilemma        | 380 |
> |Rational Coordination |  28 |
> |Stag Hunt |  35 |
> |Pure Coordination | 150 |
>
> ---
>
> **Q4: This method is computationally much more intensive than baselines.**
>
> **A4:** Our method is not computationally much more intensive than baselines. When starting a new episode, our method samples $n$ policies and loads these policies for $n$ agents to collect experiences. So our method is not computationally expensive per step. Our method uses extra memory to store the keys in the CPU’s memory and the model in the hard disk. We use 200M training steps to train all the methods. Each method uses 200M steps of experience to train the policies.
>
> ---
>
> **Q5: It is unclear how agents are chosen at evaluation time to be entered into the substrate.**
>
> **A5:** There are $n$ agents in each substrate during training. After training n policies for $n$ agents, the evaluation starts, and $m$ agents will be chosen as the background agent and loaded with pre-trained agents by Leibo et al. 2021 [2]. The rest $n-m$ agents’ policies will not be replaced.
>
> ---
>
> **Q6: The focal agent's return is clearly dependent on the co-player trained with, how do you handle this instability during training?**
>
> **A6:** We use the centralized training paradigm [3, 4, 5] to train $n$ policies for each agent introduced in Section 4.2. We use a centralized critic function in centralized training by mixing each agent's individual data [3]. The centralised critic value is used to update agents’ policies via policy gradient. The detailed architecture can be found in Appendix C.
>
> ---
>
> **Q7: The method is very similar to Fictitious Co-Play, the agents stored in the buffer are very similar to a population of agents and thus explaining the differences in related work could be useful.**
>
> **A7:** Fictitious Co-Play (FCP) aims to learn policies for a two-agent cooperative game. It is a two-stage method. In the first stage, $n$ agents are trained independently with random seeds via self-play. It needs $n$ seeds and much more extra computation. In the second stage, an FCP agent is trained by interacting with the trained policies of $n$ agents. Another extra run of training is also needed. However, our method, RPM, is an end-to-end training method for social dilemmas, competitive, cooperative, and even mixed environments where there are more than two agents. It needs only one run training by utilizing fictitious self-play to sample $n$ policies for each agent without maintaining n populations of agents.
>
> ---
>
> **Q8: No code is provided, nor is method clear enough for reproducibility.**
>
> **A8:** The code was provided, and the link can be found in Section 9 in the paper. There is a reproducibility statement in Section 9 instructing readers to use the code to reproduce the results. Network architectures and hyperparameters were also introduced in Appendix Sections C and D, respectively.

---

> > ### Author Response · Authors · 2022-11-13
> > **Response to Reviewer 8Vwu 2/2**
> >
> > **References:**
> >
> > [1] Strouse, D. J., McKee, K., Botvinick, M., Hughes, E., & Everett, R. (2021). Collaborating with humans without human data. Advances in Neural Information Processing Systems, 34, 14502-14515.
> >
> > [2] Leibo, J. Z., Dueñez-Guzman, E. A., Vezhnevets, A., Agapiou, J. P., Sunehag, P., Koster, R., ... & Graepel, T. (2021, July). Scalable evaluation of multi-agent reinforcement learning with melting pot. In International Conference on Machine Learning (pp. 6187-6199). PMLR.
> >
> > [3] Lowe, R., Wu, Y. I., Tamar, A., Harb, J., Pieter Abbeel, O., & Mordatch, I. (2017). Multi-agent actor-critic for mixed cooperative-competitive environments. Advances in neural information processing systems, 30.
> >
> > [4] Rashid, T., Samvelyan, M., Schroeder, C., Farquhar, G., Foerster, J., & Whiteson, S. (2018, July). Qmix: Monotonic value function factorisation for deep multi-agent reinforcement learning. In International conference on machine learning (pp. 4295-4304). PMLR.
> >
> > [5] Yu, C., Velu, A., Vinitsky, E., Wang, Y., Bayen, A., & Wu, Y. (2021). The surprising effectiveness of ppo in cooperative, multi-agent games. arXiv preprint arXiv:2103.01955.

---

> ### Author Response · Authors · 2022-11-17
> **Dear Reviewer 8Vwu, did our response address your concerns?**
>
> Dear Reviewer 8Vwu,
>
> We thank you again for your valuable time and comments on our paper. We made responses to your questions. Did our response address your concerns? As the response system will be closed soon within ONE day. If you have no more questions, we would appreciate it if you could kindly consider raising the score.
>
> Sincerely yours,
>
> Authors

---

### Author Response · Authors · 2022-11-13
**General Response to All Reviewers**

Dear reviewers,

We thank you all for your positive, constructive, and valuable comments. We appreciate positive comments and in-depth thoughts made by reviewers who recognized our contribution to the generalization in MARL. We briefly summarize our updates:

1. Updated the paper (updates are in blue), including:

    a. Added discussions on some related papers in Appendix F in the paper due to the 9-page limit of the main text;

    b. Included the exploiter's "optimal" performance.

2. Submitted responses as per the reviews;

We hope the responses address the concerns of all the reviewers. More discussions on our paper are also always welcome!

---

### Decision · Program_Chairs · 2023-01-20

**Decision:**

Accept: poster

**Justification For Why Not Higher Score:**

It is an interesting idea that works well in practice. The idea is intuitive and is not totally new to be considered for a spotlight.

**Justification For Why Not Lower Score:**

There is no single solid reason to reject the paper. The authors have addressed all the concerns of the reviewers and the reviewers have accepted them as well with one of them increasing their score.

**Metareview: Summary, Strengths And Weaknesses:**

This paper proposes a new MARL algorithm called RPM which maintains a memory of a diverse set of policies using the returns as keys. Experiments on the melting pot environment show that the proposed method is better than other existing algorithms.

Overall this is a solid idea with no major issues. There was one concern about the idea being a heuristic but that cannot be a reason for rejection (as the reviewer also agrees). It works well in practice and not every ML paper has a sound theory. The authors addressed all the concerns of the reviewer and added more results and related work.

I recommend an acceptance.

**Note From Pc:**

if the above contains the word "oral" or "spotlight" please see: "oral" presentation means -> notable-top-5% and "spotlight" means -> notable-top-25%. As stated in our emails, we are disassociating presentation type from AC recommendations